# Forecasting Water Temperature in Cascade Reservoir Operation-Influenced River with Machine Learning Models

**Dingguo Jiang [1], Yun Xu [2,3,*], Yang Lu [4,*], Jingyi Gao [5] and Kang Wang [1]**

1   China Three Gorges Corporation, Wuhan 430010, China; jiang_dingguo@ctg.com.cn (D.J.);
    wang_kang1998@163.com (K.W.)
2   Zhejiang Institute of Communication Co., Ltd., Hangzhou 310000, China
3   College of Water Conservancy and Hydropower Engineering, Hohai University, Nanjing 210098, China
4   Department of River and Harbor Engineering, Nanjing Hydraulic Research Institute, Nanjing 210024, China
5   School of Humanities and Social Sciences, Beijing Institute of Technology, Beijing 102488, China;
    gaojingyi@bit.edu.cn
*   Correspondence: xuyuncz@163.com (Y.X.); luyang@nhri.cn (Y.L.)

**Abstract:** Water temperature (WT) is a critical control for various physical and biochemical processes in riverine systems. Although the prediction of river water temperature has been the subject of extensive research, very few studies have examined the relative importance of elements affecting WT and how to accurately estimate WT under the effects of cascaded dams. In this study, a series of potential influencing variables, such as air temperature, dew temperature, river discharge, day of year, wind speed and precipitation, were used to forecast daily river water temperature downstream of cascaded dams. First, the permutation importance of the influencing variables was ranked in six different machine learning models, including decision tree (DT), random forest (RF), gradient boosting (GB), adaptive boosting (AB), support vector regression (SVR) and multilayer perceptron neural network (MLPNN) models. The results showed that day of year (DOY) plays the most important role in each model for the prediction of WT, followed by flow and temperature, which are two commonly important factors in unregulated rivers. Then, combinations of the three most important inputs were used to develop the most parsimonious model based on the six machine learning models, where their performance was compared according to statistical metrics. The results demonstrated that GB3 and RF3 gave the most accurate forecasts for the training dataset and the test dataset, respectively. Overall, the results showed that the machine learning model could be effectively applied to predict river water temperature under the regulation of cascaded dams.

**Keywords:** water temperature prediction; machine learning; importance ranking; air temperature; flow discharge

## 1. Introduction

Water temperature (WT) is one of the crucial factors in almost all physical and biogeochemical processes in rivers, which plays a critical role in determining the health of aquatic ecosystems [1]. For instance, high WT can promote the rapid formation of algal blooms, capable of affecting ecosystem functions and water quality [2]. Indirectly, higher WT usually result in lower dissolved oxygen in rivers, affecting aquatic organisms and a wide range of biogeochemical processes such as nitrification, denitrification and respiration [3]. Many studies have demonstrated that alterations in thermal conditions can affect aquatic organisms' life stages and growth by triggering biological behavior responses, such as fish spawning activity [4–7]. The availability of WT data at different spatial and temporal scales has improved the understanding of thermodynamics, allowing for a comprehensive review of WT processes, including the spatial and temporal variability of energy exchange [1,7–9] and the influence of large-scale drivers such as climate and flow regime [7,10] and local and regional drivers (e.g., watershed topography and water regime [11,12]). The majority

of the processes (climate and hydrology) and fluxes (radiation, latent heat, sensible heat, and advective heat exchange) that control WT are now fairly well understood [13,14].

Natural river WTs are largely determined by the heat exchange between surface air and water, as well as the temperature of the various runoff components that supply water to rivers (such as surface runoff, groundwater input, and snowmelt input) [15]. As opposed to unregulated rivers, the globally expanding anthropogenic impoundments (e.g., large dams, small reservoirs, and ponds) are affecting downstream temperature regimes in a variety of ways, depending on their structure and location [16]. Generally, prior research indicates that large dams reduce downstream temperatures by releasing cold underflow water in summer [17]. Moreover they delay the annual cycle of flow and river temperature regimes [9,18]. However, there is less information on the relative importance of the factors affecting WT and how to accurately predict WT, especially under the influence of cascaded dams. A cascade dam is a succession of reservoirs built in a stepped form from upstream to downstream of a river or river portion, and it is an important technique to develop and use a river's hydro energy resources [19].

Given the significance of temperature as a factor affecting the river environment's quality and the possibility of changes in heat state as a result of human influence [20], two distinct types of water temperature predicting models have been developed over the last few decades: physical-based and data-driven [21,22]. Physical-based models simulate the spatial and temporal variation of river water temperature based on the energy balance of heat fluxes and the mass balance of flows in the water column. These models require a large number of input variables, such as river geometry, hydrological and meteorological conditions, etc., and are, therefore, impractical and time-consuming in many cases due to their complexity. Data-driven models based on statistical or data mining techniques, on the other hand, are relatively simple, require fewer data inputs, and allow for fast predictions [23]. Machine learning models, as typical data-driven models, usually perform better for regression prediction of small volume data [24–27]. By comparing and evaluating the performance of machine learning models for predicting water temperature in previous studies, it is clear that there is no single model that can consistently outperform other models [21]. Due to the fact that the fundamental design of machine learning models varies, the WT input-output structure may not be adequately investigated in a range of scenarios, which may be improved by employing various machine learning techniques [28,29].

Jinsha River, the upstream portion of the Yangtze River, is located in the Rare and Unique Fish National Nature Reserve. It is a spawning site and important habitat for *Psephuyrus gladius, Myxocyprinus asiaticus* and other unique fishes. The development of several large cascaded hydropower dams, constructed on the Jinsha River since 2005, attracted much attention [30]. Over the past two decades, many studies have examined the impact of Jinsha River cascade reservoirs on river flow conditions, extreme drought events, sediment deposition and aquatic habitat suitability [31]. Only a few studies have looked into the connections between aquatic organisms and environmental conditions [19]. Since temperature is an important factor throughout the life cycle of aquatic organisms, there is an urgent need to accurately predict changes in WT to facilitate reservoir ladder management in the context of climate change for the benefit of biotic communities.

In this study, several machine learning models (i.e., adaptive boosting (AB), decision tree (DT), random forest (RF), support vector regression (SVR), gradient boosting (GB), and multilayer perceptron neural network (MLPNN))will be used to predict the downstream WT under the effects of the Jinsha River cascaded reservoirs. The major objectives of this paper are: (1) Assess the predictive performance of various machine learning models. (2) Identify the relative importance of each factor in predicting WT. (3) Determine the minimum factors needed to accurately predict WT. We first conducted an initial test of the performance of the six proposed models with all input factors, we then used perturbation importance to assess the relative importance of each factor, and finally, the model was simplified by testing different combinations of the most important factors. This study will

enhance the understanding of river WT under the regulation of cascaded reservoirs, and provide an optimal machine learning model for WT prediction.

## 2. Materials and Methods

### 2.1. Study Area

To address the prediction of water temperature under the cascaded reservoir, we chose the watershed where Xiangjiaba Hydropower Station is located as the study area. The dam is sited at the junction of Yibin city, Sichuan province, and Shuifu city, Yunnan province, 157 km from the Xilodu hydropower station's dam site. It controls a drainage area of 458.8 thousand km$^2$, accounting for 97 percent of the Jinsha River basin. The average annual runoff is 3810 m$^3$/s, total volume of water stored is 5.163 billion m$^3$, regulated volume is 900 million m$^3$ and the backwater length is 156.6 km [32]. Figure 1 illustrates the major hydraulic structures and hydrological stations.

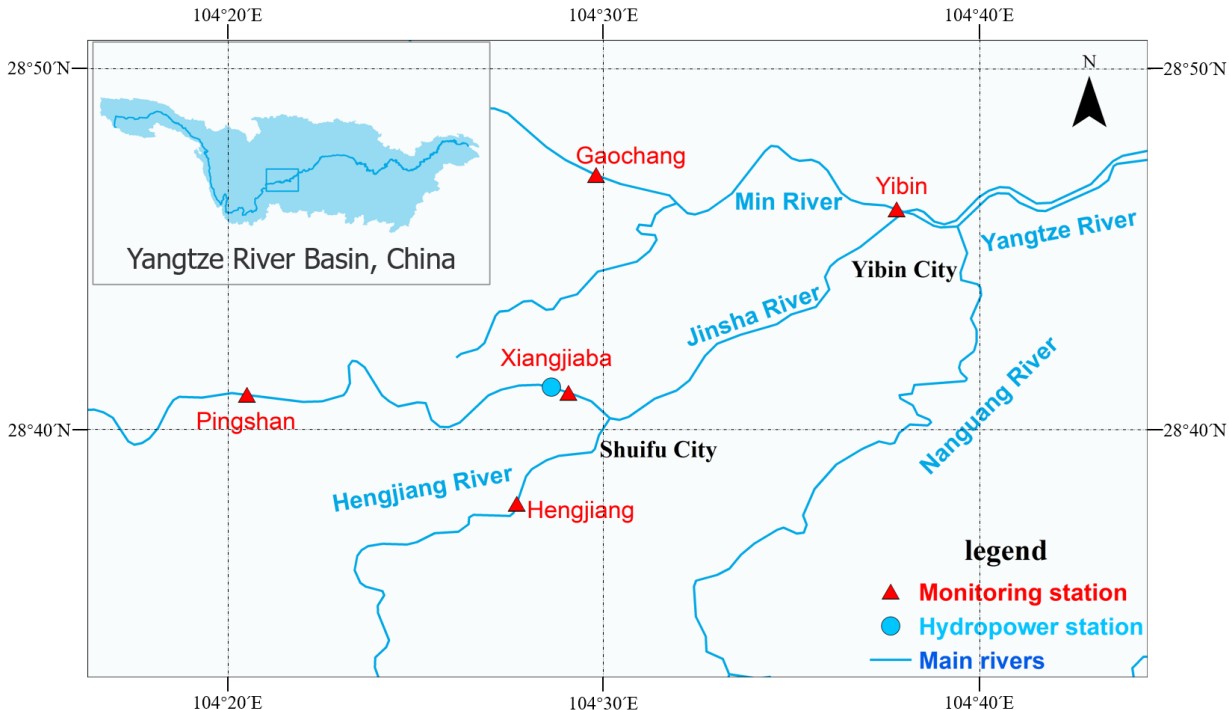

**Figure 1.** Distribution map of main dams (blue circle) and hydrological monitoring points (red triangles) in the study area.

In this study, the hydrological data are monitored and provided by Xiangjiaba and Yibin stations for the study period of 2015 to 2018. Meteorological data are sourced from the NOAA-National Centers for Environmental Information (https://www.ncei.noaa.gov/ (accessed on 15 March 2022)). We collected pertinent dew point temperature (DewT), daily average air temperature (MeanAT), maximum air temperature (MaxAT), minimum air temperature (MinAT), precipitation, average wind speed (MeanWind), and maximum wind speed (MaxWind) as predictors. The time series of annual water temperature (WaterT), averaged air temperature (MeanAT), and river flow discharge (Discharge) for the hydrological station are shown in Figure 2.

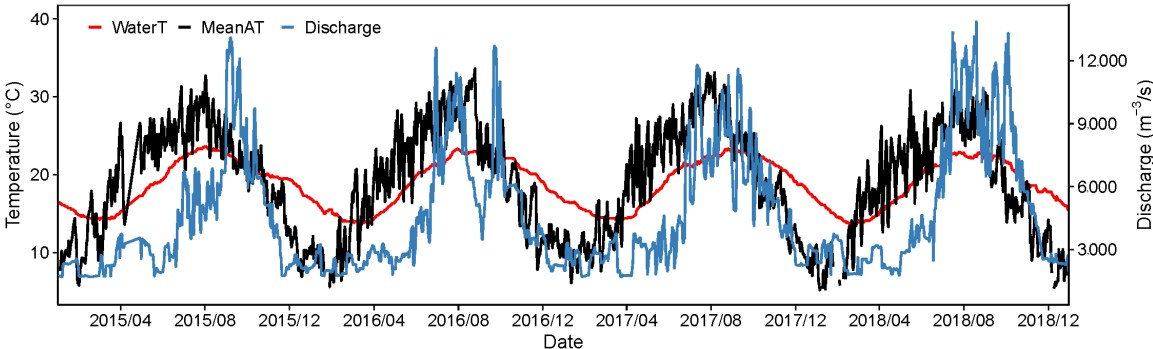

**Figure 2.** Time series plot of water temperature (red lines), mean air temperature (black lines), and discharge (blue lines)

## 2.2. Methodology

A ML model requires several steps: define the inputs (predictors), pre-processing them, and then apply the ML algorithm. All the aspects of the methodology shown in Figure 3 are analyzed individually in the following paragraphs. The six machine learning models, including Decision Trees (DT), Random Forest (RF), Gradient Boosting (GB), Adaptive Boosting (AB), Support Vector Regression (SVR) and Multilayer Perceptron Neural Network (MLPNN), are introduced in this subsection, all of the models in this study are written in Python. As usual, machine learning models, they perform well in traditional regression prediction, and we will utilize them, respectively, for sensitivity analysis and water temperature forecasting. Their types, benefits, drawbacks and applicability are detailed separately below.

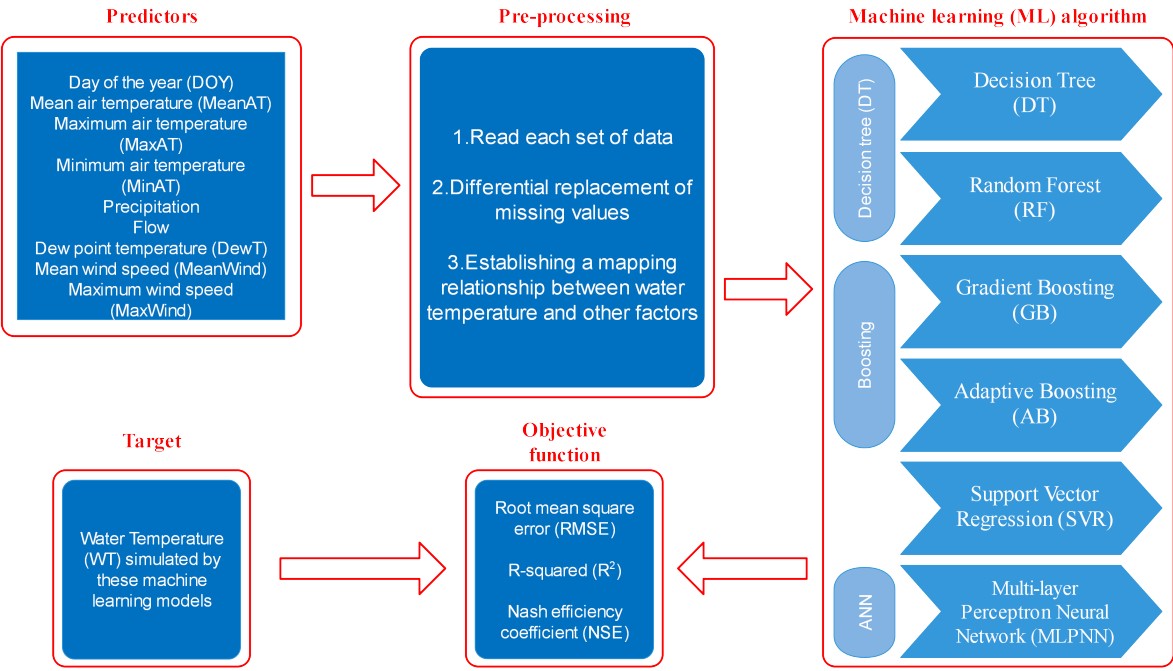

**Figure 3.** Workflow summarizing the steps of the comparative analysis of the performance of the different ML methods.

### 2.2.1. Decision Trees (DT)

For supervised learning, decision trees are a non-parametric technique. To address classification and regression problems, the synthesis decision rules from a set of attributes and labeled data and presented in the form of a tree view.

Additionally, decision trees are capable of naturally resolving multiclassification problems and have a high degree of interpretability. Ref. [33] uses a spanning tree algorithm for entropy, such as

$$I_E(i) = -\sum_{j=1}^{m} f(i,j) \log_2 f(i,j) \tag{1}$$

The advantages of decision trees include their low computational complexity, simple to understand output results, insensitivity to the absence of intermediate values, and the ability to handle unrelated feature data. The disadvantage is that there is usually a possibility of overfitting [34].

### 2.2.2. Random Forest (RF)

Random forest is a technique for ensemble supervised machine learning. It makes use of decision trees as its foundation classifier, based on the bagging principle and employs randomization in two ways: first, in the selection of training data samples for each base tree, and second, in the selection of attributes for tree induction [35]. Each decision tree evaluates and classifies a new sample independently in a classification task. If one of the decision tree's taxonomic results contains the greatest amount of taxonomy, the stochastic forest will use it as the final result [36].

There are numerous benefits to random forest: (1) Extremely high precision can be effectively applied to big datasets. (2) Incorporates randomization and so resists overfitting. (3) It can handle extremely high-dimensional data without the need for dimensionality reduction. (4) It can handle both discrete and continuous data in order to achieve parallelism without the need to normalize the data collection. (5) Even with the missing value situation, it is possible to attain positive results. Even while the technique itself is very fast, when there are a large number of decision trees in a random forest, the required training space and time are huge, resulting in a substantially slower model. In practical applications where real-time speed is crucial, it is, therefore, preferable to adopt a different strategy.

### 2.2.3. Gradient Boosting (GB)

GB is frequently used in regression trees, and its distinction from traditional boosting is that each calculation of GB is to reduce the residuals of the previous round and to eliminate the residuals, we can build the model along the gradient of residual reduction, whereas traditional boosting is primarily to weight the correct and incorrect samples. Each training round is determined by the residuals from the preceding round. In this instance, the residual is the negative gradient value of the present model. This necessitates filtering the output of the weak classifier at each iteration and solving the regression issue using the least-square error, which has the following form [37]:

$$\operatorname{argmin}_{\gamma_m} \sum_{i=1}^{N} \left[ y_i - F_{m-1}(x_i) - \sum_{j=1}^{J} \gamma_{jm} 1(x_i \in R_{jm}) \right]^2 \tag{2}$$

$$\operatorname{argmin}_{\gamma_m} \sum_{i=1}^{N} \left[ r_{mi} - \sum_{j=1}^{J} \gamma_{jm} 1(x_i \in R_{jm}) \right]^2 \tag{3}$$

where $r_m = y_i - F_{m-1}(x_i) = -\frac{1}{2} \frac{\partial L}{\partial F} \Big|_{F=F_{m-1}(x_i)}$ is the residual (negative gradient) of the ith data point. Each round of regression tree learning can be seen as a process of fitting the residual with the least-square error. The optimal value of leaf nodes is the average value of sample residuals in each leaf node:

$$\gamma_j = \operatorname{average}_{x \in R_{jm}} r_i \tag{4}$$

The advantages of the GB model are that it can easily handle a variety of different types of data and has higher prediction accuracy and shorter tuning time. The disadvantage is that there is a serial relationship between the data and the underlying learners, which makes parallel training difficult.

### 2.2.4. Adaptive Boosting (AB)

AB is a lifting algorithm that is iterative in nature. Its central concept is to train multiple classifiers (weak classifiers) on the same training set and then combine these weak classifiers to create a stronger final classifier (strong classifier) [38]. Its adaptability stems from the fact that the algorithm is implemented by altering the data distribution. It weights each sample based on the accuracy of the classification of each sample in each training set and the accuracy of the most recent overall classification. The new dataset with modified weights is then used to train the lower classifier. Finally, the final decision classifier is constructed by fusing the classifiers obtained from each training.

AB has the advantage of treating several classification techniques as weak classifiers while taking into account the weight of each classifier. The downside is that the number of repetitions of AB, i.e., the number of weak classifiers, is not adequately stated. The resulting data imbalance results in decreased classification accuracy and a longer training period.

### 2.2.5. Support Vector Regression (SVR)

Support vector machine regression (SVR) is a model for the application of support vector machines (SVM) to regression problems. It works by finding a hyperplane such that the distance between all data points and this hyperplane is minimized [39,40]. Using SVR for regression analysis, like SVM, we need to find a hyperplane. The difference is, in SVM, we need to find a hyperplane with the largest gap, while in SVR, we define a hyperplane $\epsilon$, the residual of the data points defining the area within the target is 0, and the distance from the data points (support vectors) outside the region to the boundary is the residual ($\zeta$). Similar to the linear model, we want these residuals ($\zeta$) to be minimal. Therefore, in general, SVR is about finding the optimal band region ($2\epsilon$ width) and then regressing the points outside that region.

SVR has the following characteristics: it is adaptable, and works well with low and high-dimensional data. When dealing with large samples, data pre-processing and tuning are required, or the algorithm will perform poorly [41].

### 2.2.6. MultiLayer Perceptron Neural Network (MLPNN)

The MLPNN model is well-known for its ability to rapidly train and learn the relationship between a set of input variables and an output parameter. Using a subset of the input dataset, MLPNN can determine the optimal mathematical relationship for predicting an outcome. Then, using the statistical values, the generated error can be calculated [42]. The model can be trained in three stages: feed-forward, error calculation, and backward circulation. Feed-forwarding is a technique in which the model passes the input data to middle layers that multiply it with randomly chosen weights, adds biases at each layer, and computes the model's predicted output. Following that, the difference between the predicted and experimental values is calculated and back-propagated to update the weights and biases, which are repeated until the error is minimized. The MLPNN regressor will have a high recognition rate and will perform well in terms of regression speed. In exchange, training takes longer than SVR, particularly for large training sets.

## 3. Results

### 3.1. Importance Ranking of Variables

Prior to selecting the most essential features to be integrated as initial model inputs, it is necessary to examine the variables' relative importance. Six machine learning models, including decision trees (DT), random forests (RF), gradient boosting regression (GB), adaptive boosting regression (AB), support vector regression (SVR) and multilayer perceptron

neural networks (MLPNN), were used to determine the relative importance of each factor for predicting accuracy in this subsection. All six models achieved good performance in the training data, as shown in Table 1, with their root mean square error (RMSE) no more than 0.34 °C, $R^2$ all above 0.96 and Nash efficiency coefficient (NSE) [43] no less than 0.96; the best-fitting model was GB, which was almost completely accurate for all training data. In the test data, the prediction accuracy of the six models decreases to some extent. However, the overall RMSE of all models was below 0.64 °C, the $R^2$ was above 0.93, and the NSE was no less than 0.92. The RF model achieved the highest precision among all of them, with an RMSE of only 0.203 °C. All of the above indicate that the constructed model is robust in terms of variable screening.

**Table 1.** Performances of six models (DT: decision trees, RF: random forests, GB: gradient boosting regression, AB: adaptive boosting regression, SVR: support vector regression, MLPNN: multilayer perceptron neural networks) in predicting water temperature.

| Models | Training Datasets | | | Testing Datasets | | |
|---|---|---|---|---|---|---|
| | RMSE (°C) | $R^2$ | NSE | RMSE (°C) | $R^2$ | NSE |
| DT | 0.0167698 | 0.998298 | 0.998295 | 0.394039 | 0.959443 | 0.959352 |
| RF | 0.0513479 | 0.994788 | 0.994694 | 0.203134 | 0.979092 | 0.978487 |
| GB | $2.59 \times 10^{-19}$ | 1 | 1 | 0.308065 | 0.968292 | 0.968119 |
| AB | 0.0980462 | 0.990049 | 0.989885 | 0.264675 | 0.972758 | 0.972145 |
| SVR | 0.3365923 | 0.965837 | 0.960535 | 0.633647 | 0.934781 | 0.922508 |
| MLPNN | 0.1896209 | 0.980754 | 0.980438 | 0.385593 | 0.960312 | 0.958261 |

We evaluated the significance of the variable based on the significance of the perturbation. The permutation significance is defined as the difference between the variable's baseline metric and the permutation metric. We utilized root mean square error as a metric (RMSE). Each model utilized 60% of the input for training purposes and 40% for testing. As input variables to a machine learning model can directly influence the quality of the model, we incorporated all the collected data: air temperature (MinAT, MaxAT, MinAT), flow, dew point temperature (DewT), precipitation, wind speed (MeanWind, MaxWind) and day of the year (DOY). When using DOY, as its behavior at the end of each year is discontinuous, i.e., from day 1 to day 365, regression continuously changes. Therefore the $\sin(DOY/n_{DOY})$ and $\cos(DOY/n_{DOY})$ were added, where $n_{DOY}$ is the number of days in a year, these are considered two different input variables.

In the training dataset for DT, as shown at the top of Figure 4, DOY took up the value between 3.5 and 4 °C, flow weighting value of 0.9 °C ranked second, MinAT ranked third, and MeanAT, MeanWind, DewT, MaxAT and MaxWind all had an effect on the results with weight values between 0.1 and 0.4 °C. DT produced nearly identical results in the training and test datasets, with the exception of a slight increase in the test dataset's importance of MaxWind.

As for RF, the bottom part of Figure 4 shows the importance of not being the same: DOY has a weight score of 3.4 °C, flow has a weight score of 0.9 °C and DewT, MinAT and Mean wind speed all have weight scores between 0.1 and 0.3 °C. MeanAT, MaxAT, MaxWind and Precipitation all had a negligible effect, whereas SDOY and CDOY had no effect. It gave essentially equal findings in the training and test datasets, except for a modest increase in the relevance of flow in the test dataset.

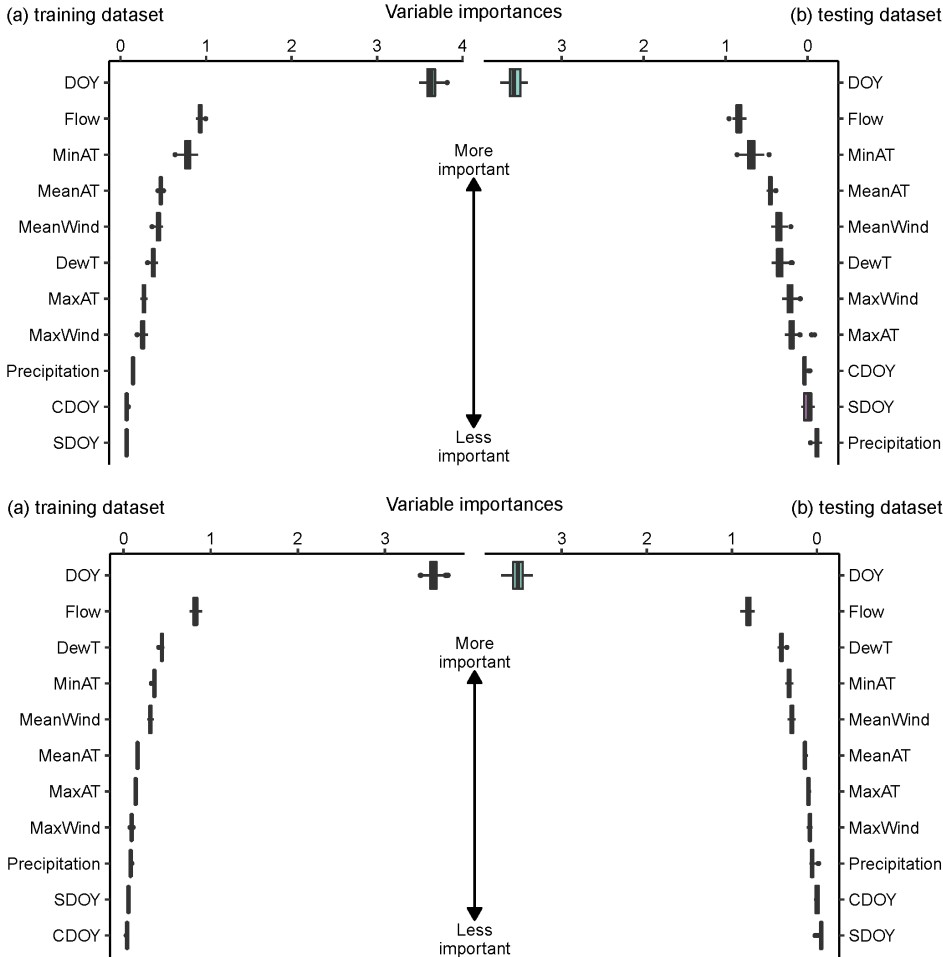

**Figure 4.** Permutation importance in DT and RF; DT, decision trees, RF, random forests. (DT on the **top**, RF on the **bottom**, WT: °C)

In the GB training dataset, which is depicted at the top of Figure 5, DOY had a weight of 3.7 °C, making it the most influential predictor. The second most important factor was river flow, which had a weight value close to one. Mean wind speed, MinAT and DewT also had an effect on the results, with their respective weight values ranging from 0.1 to 0.3 °C, while other factors, such as MaxAT, MaxWind and so on, had little effect. The order of importance of the factors remains largely unchanged in the test dataset. It is worth mentioning that SDOY, CDOY and precipitation did not work in either set.

Regarding the model AB depicted in Figure 5, we discovered that the importance of each factor was similar to that of the GB. The DOY value of 3.6 °C remained the critical element as in the previous model; flow was the second significant factor in predicting WT. In comparison to GB, a significant difference is that MinAT has reduced importance, now ranking fourth, taking its place is Mean Wind. Precipitation was slightly more important in the test dataset than in the training dataset; all other factors were similar.

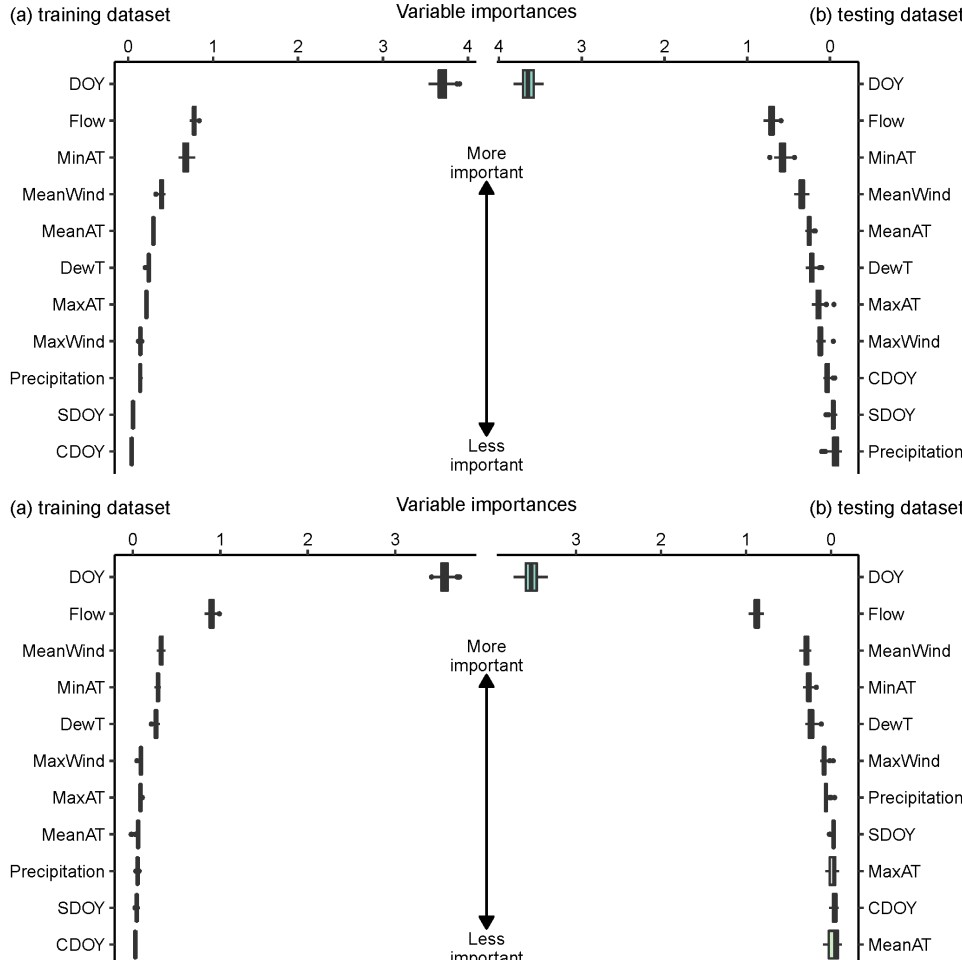

**Figure 5.** Permutation importance in GB and AB; GB, gradient boosting regression, AB, adaptive boosting regression. (GB on the **top**, AB on the **bottom**, WT: °C)

In the training dataset, SVR produced very different results than the previous ones, as illustrated in the upper part of Figure 6, with the weight value of DOY falling below 3.0 °C for the first time. DOY and flow were nearly equally important in the SVR model. In other words, flow was a significant predictor of WT in SVR. Correspondingly, the proportion of variables in the second tier decreased, i.e., the weight values of MaxAT, DewT, MeanAT and MinAT retained a low degree of consistency, and the remaining factors did not play a role. This result was also reflected in the test dataset, where DOY and Flow had weight values of approximately 2.5 and 2.2 °C, with DOY being the dominant factor. All other significance values are consistent with those found in the training dataset.

The MLPNN model presents similar results in the training and test datasets, with DOY ranked first, MeanAT in second place, DewT, MinAT, Flow, and MaxAT having similar weight values in the third echelon, and the other factors had little or no effect. The specific situation is shown in Figure 6.

We ranked the input variables in order to identify the WT predictors of interest for the next step of modeling. For the results of importance ranking, we discovered that DOY and its variation were significantly more important than AT and flow, which were typically believed to be the dominant factors.

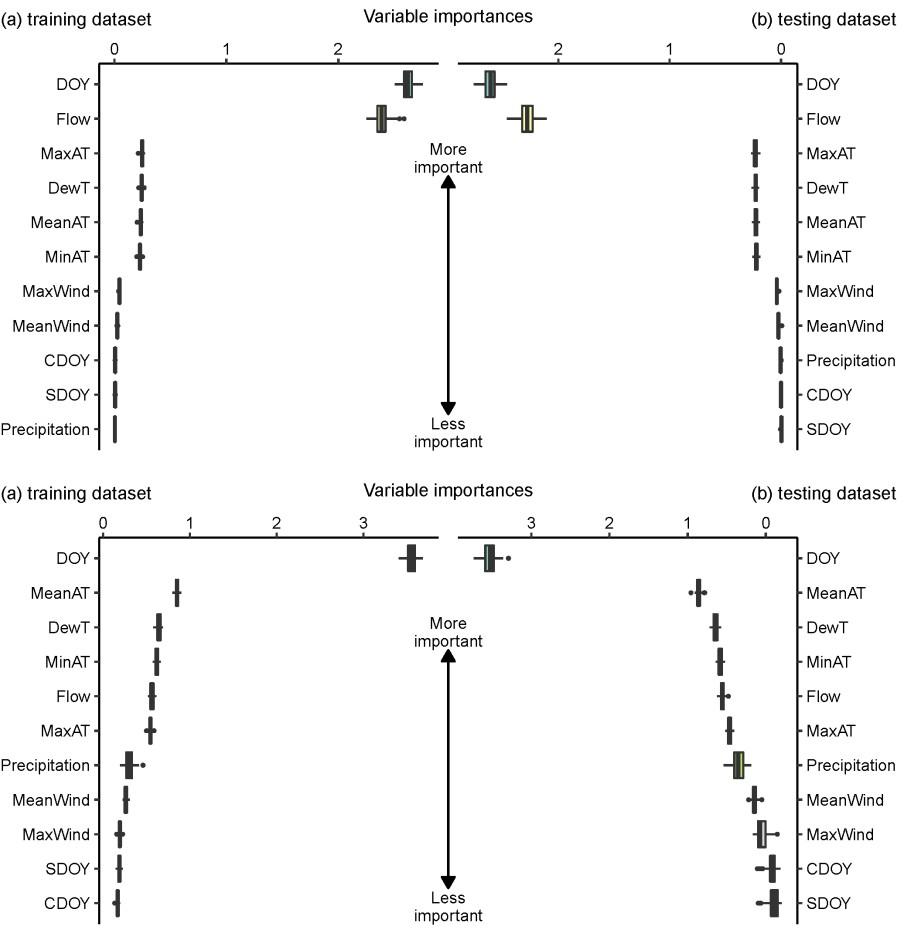

**Figure 6.** Permutation importance in SVR and MLPNN; SVR, support vector regression, MLPNN, multilayer perceptron neural networks. (SVR on the **top**, MLPNN on the **bottom**, WT: °C)

As a result of this subsection's conclusion that DOY and Flow are frequently significant factors in predicting water temperature under the influence of cascaded reservoirs, the following subsection developed six models in accordance with the following three different versions: (a) Version 1 used a single input variable (DOY), (b) Version 2 used two input variables (DOY and Flow), and (c) Version 3 used all variables. The three versions of each machine learning model are based on the same structure (i.e., AB, DT, RF, etc.), while they represent three significantly different river temperature models due to the different combinations of predictor variables.

### 3.2. Prediction Results of Each Model

In this subsection, we separately evaluated the performance of the three versions in six machine learning methods, and test the effectiveness of different models in forecasting water temperature under these combinations. The three statistical indices, RMSE (°C), $R^2$ and NSE, were used to compare the various machine learning models. To calibrate and validate the models used for water temperature prediction, the data were randomly divided into a calibration and validation period at a ratio of 6:4. That is, we use the first 60% of the data as input to train the models and then use the remaining 40% to test the model's performance. This is a widely used paradigm for model development.

The performance of DT in predicting WT is summarized in Table 2. The RMSE and $R^2$ values for each model version in the training set are 0.200 °C and 0.980, 0.044 °C and 0.996 and 0.017 °C and 0.998, respectively. While the RMSE and $R^2$ values for each model version in the test dataset are 0.554 °C and 0.943, 0.330 °C and 0.966 and 0.359 °C and 0.963, respectively. Specifically, DT achieves greater accuracy in versions 2 and 3, both of which

improve by approximately two percentage points over version 1. This demonstrates that including Flow as a predictor aids in reducing forecast error. Switching from version 2 to version 3 and including factors other than DOY and Flow as input variables improved the performance of DT slightly. The scatterplot and comparison between observed and predicted WT for the three-model version are shown in Figure 7. As illustrated in the figure, the graphs for all three cases fit reasonably well, and the scatter on the regression curves is relatively concentrated.

Table 3 summarizes the results for three different versions of RF in predicting WT. The results indicate that models developed with only the DOY as an input variable had a high correlation with the observed WT, as indicated by the high $R^2$ and NSE values ($R^2 \approx 0.950$, NSE $\approx 0.949$) and a low error value: RMSE $\approx 0.201$ °C. In addition, the modeling findings reveal that version 3 greatly outperformed the other models. In fact, RF2 increased the model's accuracy by reducing the RMSE of RF1 by 49.73 percent, while RF3 enhanced the model's accuracy by reducing the RMSE by about 74.39 percent. This proportion is altered to 51.52 percent and 58.12 percent in the test set, demonstrating that the RF3 and RF2 models perform comparably. The accuracy of the three models as a whole demonstrates that version 3 outperforms all other models. Figure 8 illustrates the scatterplots and comparison of observed and predicted WT.

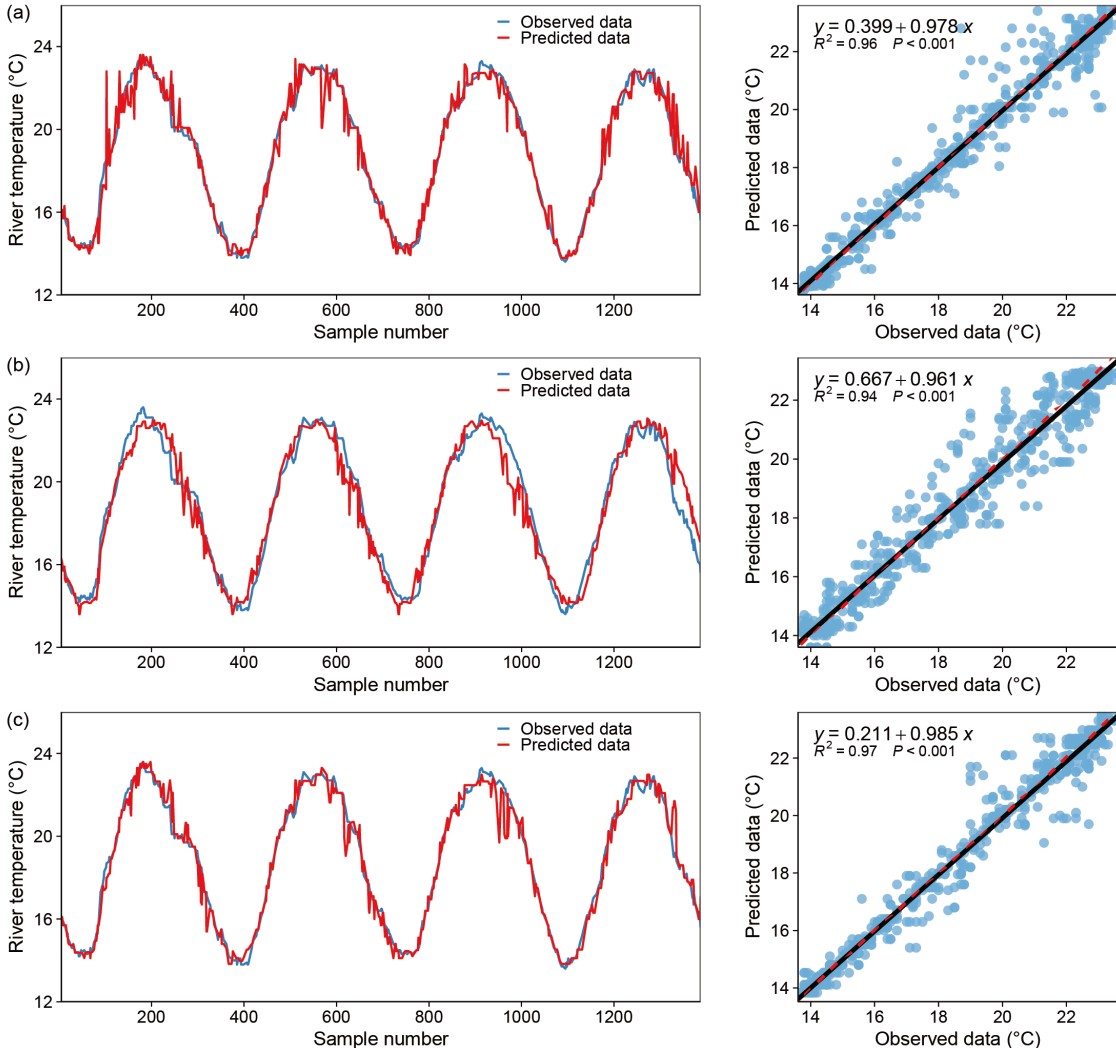

**Figure 7.** Model fitting results—DecisionTree Regressor, blue dot: X coordinate (observed data), Y coordinate (predicted data) ; black line: y = x ; red dotted line: the regression curve of the blue dots. (**a**) only one input variable (DOY), (**b**) two input variables (DOY and Flow), (**c**) all variables.

**Table 2.** Performances of DecisionTree Regressor in modeling water temperature (WT: °C), DT1: only one input variable (DOY), DT2: with two input variables (DOY and Flow), DT3: with all input variables.

| Model Version | Training Dataset | | | Test Dataset | | |
|---|---|---|---|---|---|---|
| | **RMSE (°C)** | **R²** | **NSE** | **RMSE (°C)** | **R²** | **NSE** |
| DT1 | 0.200361 | 0.979664 | 0.979242 | 0.553881 | 0.942991 | 0.941738 |
| DT2 | 0.043619 | 0.995573 | 0.995553 | 0.329739 | 0.966061 | 0.966209 |
| DT3 | 0.01677 | 0.998298 | 0.998295 | 0.359478 | 0.963 | 0.962799 |

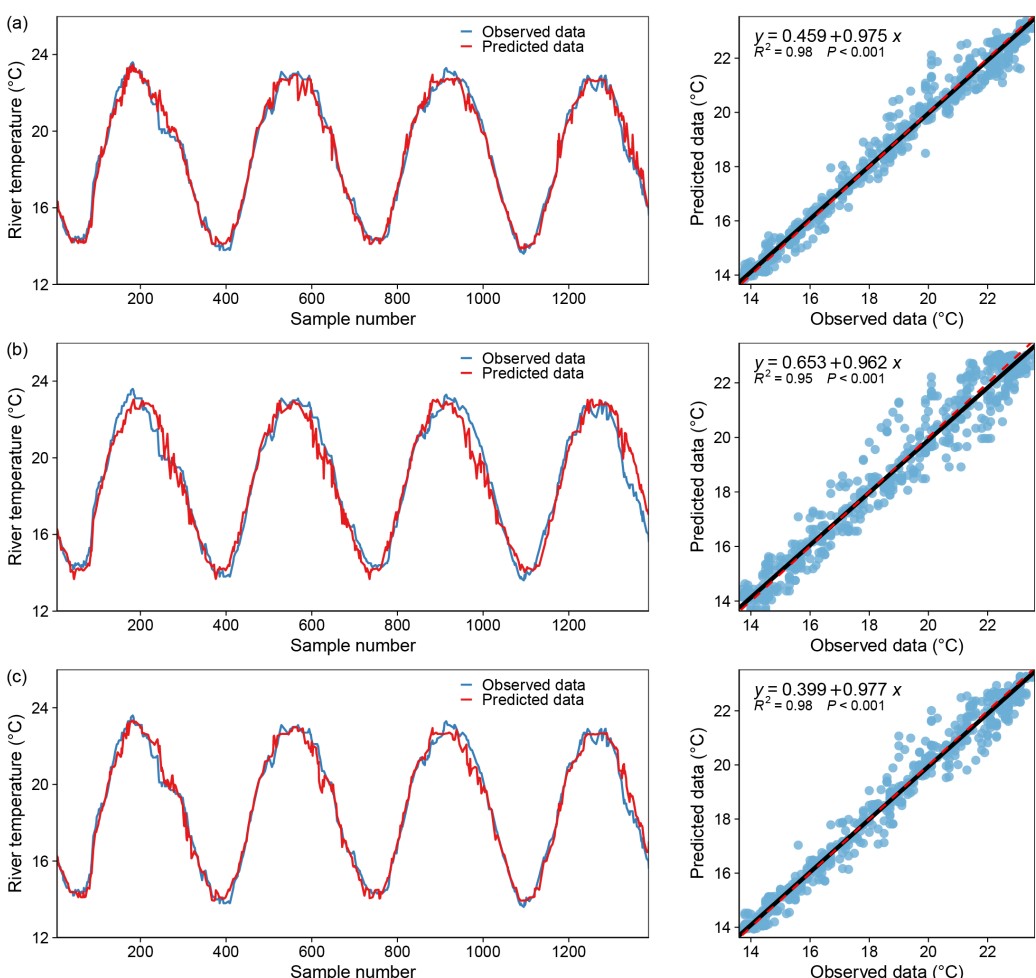

**Figure 8.** Model fitting results—RandomForest Regressor, blue dot: X coordinate (observed data), Y coordinate (predicted data) ; black line: y = x ; red dotted line: the regression curve of the blue dots. (**a**) only one input variable (DOY), (**b**) with two input variables (DOY and Flow), (**c**) with all variables.

**Table 3.** Performances of RandomForest Regressor in modeling water temperature (WT: °C), RF1: only one input variable (DOY), RF2: with two input variables (DOY and Flow), RF3: with all input variables.

| Model Version | Training Dataset | | | Test Dataset | | |
|---|---|---|---|---|---|---|
| | **RMSE (°C)** | **R²** | **NSE** | **RMSE (°C)** | **R²** | **NSE** |
| RF1 | 0.200538 | 0.979646 | 0.979199 | 0.485061 | 0.950074 | 0.948793 |
| RF2 | 0.100807 | 0.989769 | 0.989547 | 0.23512 | 0.9758 | 0.975281 |
| RF3 | 0.051348 | 0.994788 | 0.994694 | 0.203134 | 0.979092 | 0.978487 |

Table 4 presents the findings of three variants of the GB model for predicting WT. The results indicate that models developed with only the DOY as an input variable had a high correlation with the observed WT, as indicated by the high $R^2$ and NSE values ($R^2 \approx 0.980$, NSE $\approx 0.980$) and a low error value: RMSE $\approx 0.194$ °C. Additionally, the modeling results indicate that version 3 outperformed the previous versions significantly. Indeed, GB2 reduced the root mean square error of the GB1 by 99.69 percent, and GB3 increased the model's accuracy by reducing the root mean square error by approximately 100.0 percent in the training set. This percentage is changed to 48.28 and 47.32 percent in the test set, indicating a small performance difference between the GB3 and GB2 models. The accuracy of the three models as a whole demonstrates that version 3 outperforms all other models; Figure 9 illustrates the scatterplots and comparison of observed and predicted WT.

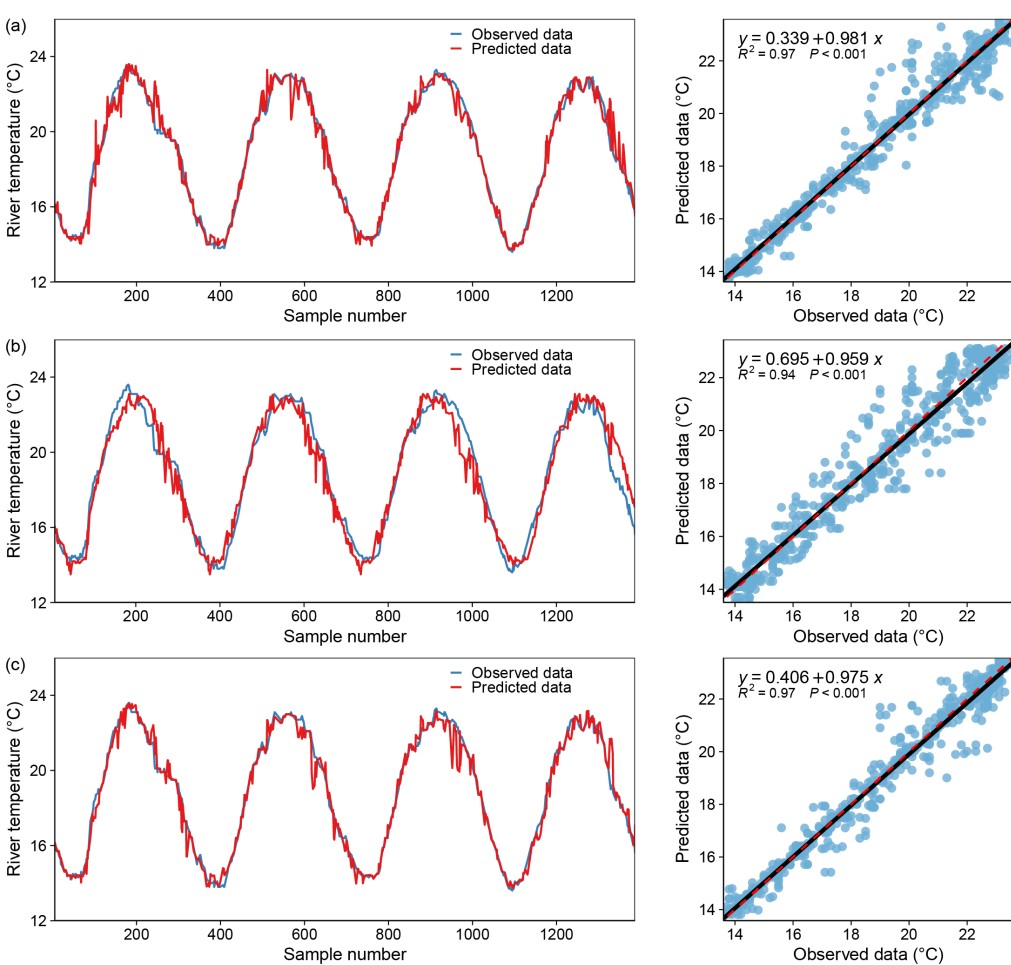

**Figure 9.** Model fitting results—GradientBoosting Regressor, blue dot: X coordinate (observed data), Y coordinate (predicted data) ; black line: y = x ; red dotted line: the regression curve of the blue dots. (**a**) only one input variable (DOY), (**b**) with two input variables (DOY and Flow), (**c**) with all variables.

**Table 4.** Performances of GradientBoosting Regressor in modeling water temperature (WT: °C), GB1: only one input variable (DOY), GB2: with two input variables (DOY and Flow), GB3: with all inputs variable.

| Model Version | Training Dataset | | | Test Dataset | | |
|---|---|---|---|---|---|---|
| | RMSE (°C) | $R^2$ | NSE | RMSE (°C) | $R^2$ | NSE |
| GB1 | 0.194251 | 0.980284 | 0.979888 | 0.584754 | 0.939813 | 0.938425 |
| GB2 | 0.000596 | 0.99994 | 0.999939 | 0.302442 | 0.968871 | 0.968273 |
| GB3 | $2.59 \times 10^{-19}$ | 1 | 1 | 0.308065 | 0.968292 | 0.968119 |

The training and test results for model AB are detailed in Table 5, indicating that the daily WT calculated using DOY and Flow is in good agreement with the observed values. As illustrated in Figure 10, when all inputs are utilized as predictors, the accuracy of the models is only marginally superior to version 1 in the training dataset and almost identical in the test dataset, indicating that the contribution of the flow is roughly equivalent to that of the other noise effects. DOY has a root mean square error of 0.203 °C and an $R^2$ value of 0.978 while operating alone (version 1) in the exercise set. When the Flow is added to DOY (version 2), the model's performance is improved slightly. In the test set, a similar phenomenon is observed, but with a lower RMSE and a higher $R^2$ value (RMSE = 0.292 °C, $R^2$ = 0.970) for version 2.

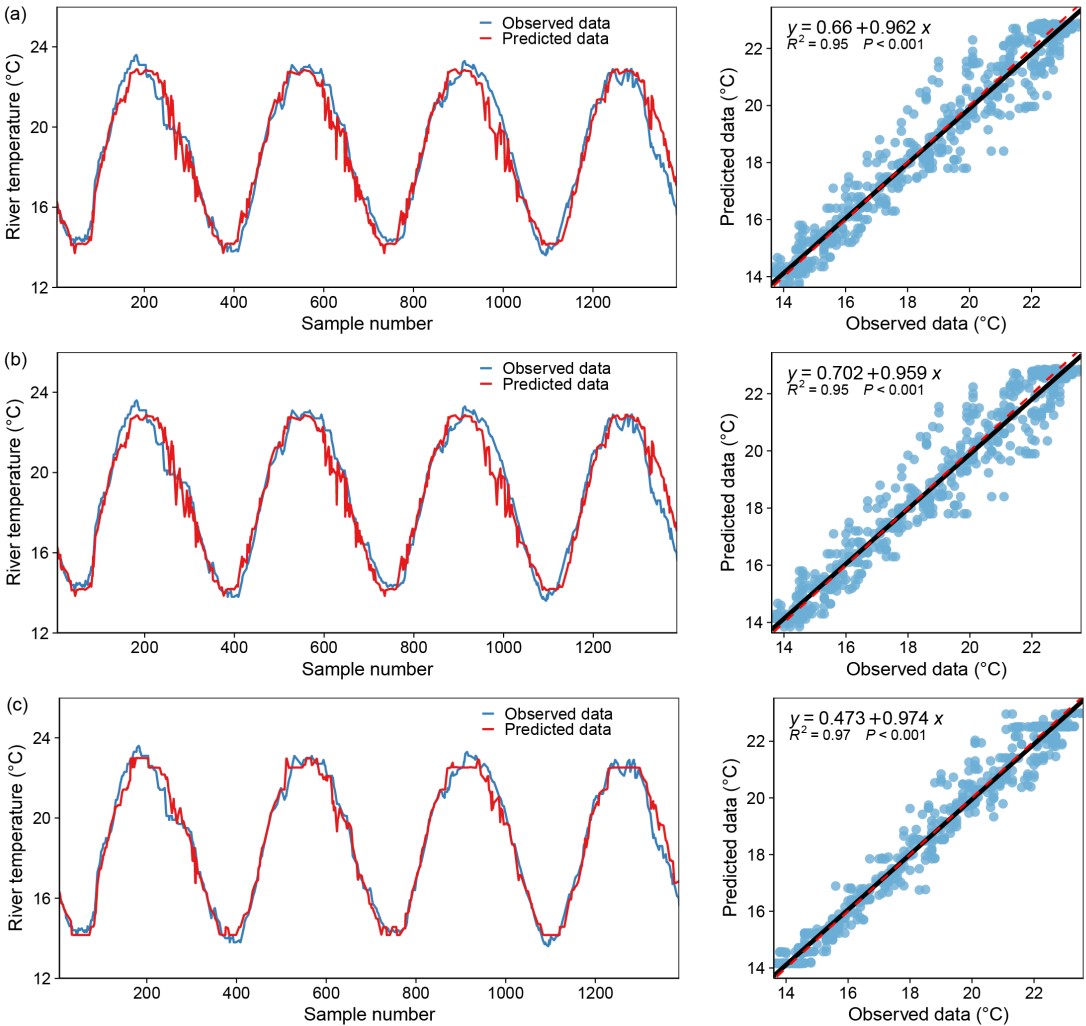

**Figure 10.** Model fitting results—AdaptiveBoosting Regressor, blue dot: X coordinate (observed data), Y coordinate (predicted data) ; black line: y = x ; red dotted line: the regression curve of the blue dots. (**a**) only one input variable (DOY), (**b**) with two input variables (DOY and Flow), (**c**) with all variables.

The detailed training and test results for SVR are shown in Table 6, which indicates that the daily WT calculated solely using DOY is in good agreement with the observed values. The scatterplots and comparisons of observed and predicted WT are shown in Figure 11. For version 2 with DOY and Flow as predictors, and for version 3 with all inputs as predictors, the accuracy of these two models is slightly lower than version 1 in the training dataset and almost identical in the test dataset, indicating that the input of flow and other factors contributes some noise interference. When DOY acts alone (version 1) in the

exercise set, it has an RMSE of 0.288 °C and an $R^2$ value of 0.971. When the Flow is added to DOY (version 2), the model's performance slightly degrades, with the RMSE increasing to 0.345 °C and the $R^2$ value decreasing to 0.965. In the test set, a similar phenomenon is observed, but with a lower RMSE and a higher $R^2$ value (RMSE = 0.323°C, $R^2$ = 0.967) for version 1.

**Table 5.** Performances of AdaptiveBoosting Regressor in modeling water temperature (WT: °C), AB1: only one input variable (DOY), AB2: with two input variables (DOY and Flow), AB3: with all inputs variable.

| Model Version | Training Dataset | | | Test Dataset | | |
|---|---|---|---|---|---|---|
| | RMSE (°C) | $R^2$ | NSE | RMSE (°C) | $R^2$ | NSE |
| AB1 | 0.202579 | 0.979439 | 0.978967 | 0.535933 | 0.944838 | 0.943332 |
| AB2 | 0.15983 | 0.983778 | 0.983445 | 0.291648 | 0.969982 | 0.969293 |
| AB3 | 0.20119 | 0.97958 | 0.979129 | 0.538943 | 0.944528 | 0.943291 |

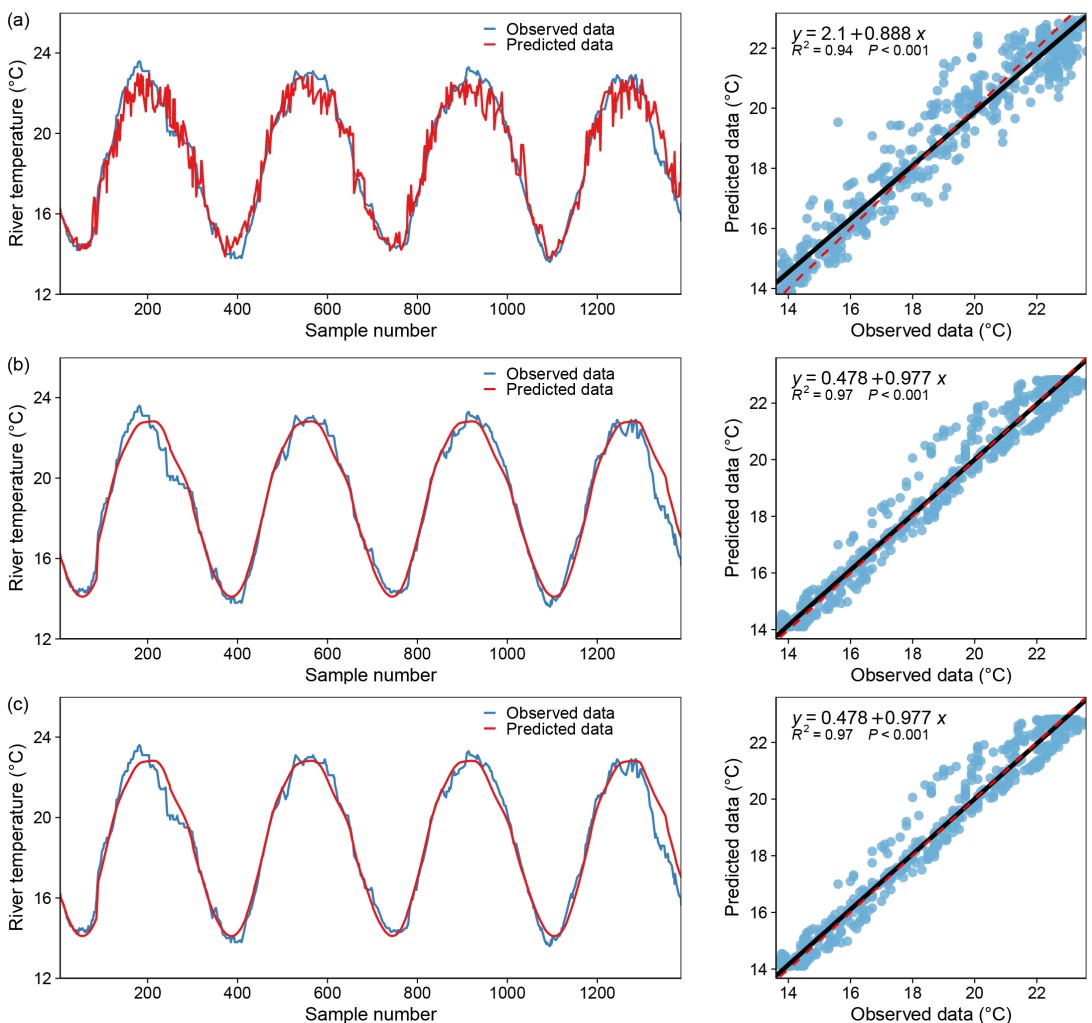

**Figure 11.** Model fitting results—SupportVector Regression, blue dot: X coordinate (observed data), Y coordinate (predicted data) ; black line: y = x ; red dotted line: the regression curve of the blue dots. (**a**) only one input variable (DOY), (**b**) with two input variables (DOY and Flow), (**c**) with all variables.

**Table 6.** Performances of SupportVector Regression in modeling water temperature (WT: °C), SVR1: only one input variable (DOY), SVR2: with two input variables (DOY and Flow), SVR3: with all input variables.

| Model Version | Training Dataset | | | Test Dataset | | |
|---|---|---|---|---|---|---|
| | RMSE (°C) | $R^2$ | NSE | RMSE (°C) | $R^2$ | NSE |
| SVR1 | 0.288101 | 0.970759 | 0.970421 | 0.323232 | 0.966731 | 0.966283 |
| SVR2 | 0.34541 | 0.964942 | 0.959725 | 0.630718 | 0.935082 | 0.923919 |
| SVR3 | 0.336592 | 0.965837 | 0.960535 | 0.633647 | 0.934781 | 0.922508 |

The performance of MLPNN in predicting WT is summarized in Table 7. The RMSE and $R^2$ values for each model version in the training set are 0.275 °C and 0.972, 0.236 °C and 0.976 and 0.190 °C and 0.981, respectively. While the RMSE and $R^2$ values for each model version are 0.311 °C and 0.968, 0.281 °C and 0.971 and 0.386 °C and 0.960, respectively, in the test dataset. It is clear that MLPNN performs better in version 2 than in version 1, with both versions improving by approximately 0.5 percentage points over version 1 in both datasets. This demonstrates that including Flow as a predictor aids in reducing forecast error. Switching from version 2 to version 3, with input variables other than DOY and traffic, improves accuracy slightly on the training set while decreasing it slightly on the test set. As illustrated in Figure 11, the graphs for all three cases fit reasonably well, and the scatter on the regression curves is relatively concentrated.

**Table 7.** Performances of Multilayer Perceptron Neural Network in modeling water temperature (WT: °C), MLPNN1: only one input variable (DOY), MLPNN2: with two input variables (DOY and Flow), MLPNN3: with all input variables.

| Model Version | Training Dataset | | | Test Dataset | | |
|---|---|---|---|---|---|---|
| | RMSE (°C) | $R^2$ | NSE | RMSE (°C) | $R^2$ | NSE |
| MLPNN1 | 0.274755 | 0.972113 | 0.970939 | 0.311318 | 0.967957 | 0.965799 |
| MLPNN2 | 0.236238 | 0.976023 | 0.975322 | 0.280818 | 0.971096 | 0.969259 |
| MLPNN3 | 0.189621 | 0.980754 | 0.980438 | 0.385593 | 0.960312 | 0.958261 |

The results for the best version of the model's performance in predicting water temperature (WT: °C) are shown in Table 8. The comparisons within each machine learning model revealed that version 3 (i.e., the model with all inputs as predictors) of the four developed models (DT, RF, GB and MLPNN) achieved the highest accuracy on the training dataset and outperformed the other versions in terms of higher $R^2$ and NSE values and lower RMSE (°C) values. The AB model performed best in version 2 (when DOY and Flow were used as predictors), while the model SVR performed better in version 1 (with DOY as an input variable only). On the test dataset, the six models produced results that were inconsistent with those obtained on the training set. Version 2 (with combined DOY and Flow inputs) had the highest fitting accuracy for models AB, DT, GB and MLPNN, while version 3 had the best result for RF and version 1 had the best result for model SVR.

Comparing the six developed models, GB achieved the highest accuracy in training data and outperformed all other models in terms of higher $R^2$ and NSE values as well as lower RMSE values. On average, the six models performed well in predicting river water temperature, with an $R^2$ value greater than 0.95 and an NSE value greater than 0.95. The DT and GB models fit curves more precisely than the others, especially at peaks and troughs. Their performance varies significantly in the test data. RF performed well in forecasting WT, with an $R^2$ greater than 0.97; AB, DT, GB and MLPNN models performed slightly worse but still exceeded 0.96; and SVR performed poorly in both cases. One possible explanation is that the data pre-processing and parameter selection algorithms

(normalization range, dimension reduction algorithm and optimal parameter selection algorithm) are not well adjusted.

**Table 8.** Best version of performance in different models (DT: decision trees, RF: random forests, GB: gradient boosting regression, AB: adaptive boosting regression, SVR: support vector regression, and MLPNN: multilayer perceptron neural networks) in predicting water temperature (WT: °C), 1.only one input variable (DOY), 2. with two input variables (DOY and Flow), 3. with all variables.

| | Training Dataset | | | | Test Dataset | | |
| --- | --- | --- | --- | --- | --- | --- | --- |
| **Models** | **RMSE (°C)** | $R^2$ | **NSE** | **Models** | **RMSE (°C)** | $R^2$ | **NSE** |
| DT3 | 0.017 | 0.998 | 0.998 | DT2 | 0.330 | 0.966 | 0.966 |
| RF3 | 0.051 | 0.995 | 0.995 | RF3 | 0.203 | 0.979 | 0.978 |
| GB3 | $2.6 \times 10^{-19}$ | 1.000 | 1.000 | GB2 | 0.302 | 0.969 | 0.968 |
| AB2 | 0.160 | 0.984 | 0.983 | AB2 | 0.292 | 0.970 | 0.969 |
| SVR1 | 0.288 | 0.971 | 0.910 | SVR1 | 0.323 | 0.967 | 0.966 |
| MLPNN3 | 0.190 | 0.981 | 0.980 | MLPNN2 | 0.281 | 0.971 | 0.969 |

Generally, and consistent with prior research, AT, flow, and DewT were all linked with changes in WT, with flow playing a more significant influence than the other two. Flow has an effect on heat exchange in rivers, and many bodies of water display high hysteresis, which obscures the link between water temperature and air temperature. The DOY is the most important element in all forecast models, suggesting that the seasonal changes in river water temperature caused by cascaded reservoirs are notable. The hydrological station utilized in this study is placed near the reservoir's downstream end, indicating that the operation regulations for cascaded reservoirs may play a substantial role in regulating the downstream rivers.

## 4. Discussion

The Jinsha River, in the upper reaches of the Yangtze River, is an important spawning site and habitat for fish, and since 2005, several large cascaded hydropower dams have been built on the Jinsha River [32]. The construction of the dams affected the water temperature, and the change in water temperature disrupted the spawning time of the fish, affecting the survival of the fish after spawning and even leading to the endangerment of some fish [17,44,45]. Accurate prediction of water temperature during the fish spawning season can help modify reservoir operation modes and the depth of water release, thereby better balancing ecological and socio-economic demands. Compared to physical-based models, we selected machine learning models that require fewer input variables to predict water temperature. The importance ranking of variables was also performed to quantify the respective contributions of these factors to the downstream thermal system. The ranking results provided a view to understand inghow the construction and operation of cascaded reservoirs affect the water temperature of downstream rivers. In addition, the ranking results can be used to obtain acceptable predicting results with minimum factors that contain major information on water temperature variations.

According to Table 1, the predictive performance of all six machine learning models (Decision Trees (DT), Random Forest (RF), Gradient Boosting (GB), Adaptive Boosting (AB), Support Vector Regression (SVR) and Multilayer Perceptron Neural Network (MLPNN)) are great, except for SVR, which has slightly worse testing results (RMSE = 0.63 °C, $R^2$ = 0.93, NSE = 0.92). Eventually, we did not find a definite answer about a single optimal machine learning algorithm when using the same input variables, indicating that our selected machine learning models all capture the nonlinear dynamics of the water temperature fairly well. The most parsimonious models were then developed based on six machine learning models using a combination of the three most important inputs. Comparing their performance according to statistical metrics, the results showed that GB3 and RF3 produce the highest prediction accuracy on the training dataset and the test dataset, respectively

(Table 8). This also suggested that choosing the appropriate minimum number of input variables is sufficient for the machine learning model to obtain acceptable prediction results. On the other hand, we can see from Figures 7–12 that as the water temperature increases (>18 °C), more discrete points appear near the fitting curve. The challenge here is that 18 °C is a threshold of great importance for fish spawning. The four major carp species only started spawning when the water temperature increased to 18 °C in April [46]. The results suggested that gaining further insight into physical dynamics remains the most essential factor for the successful exploitation of ML to better predict water temperature.

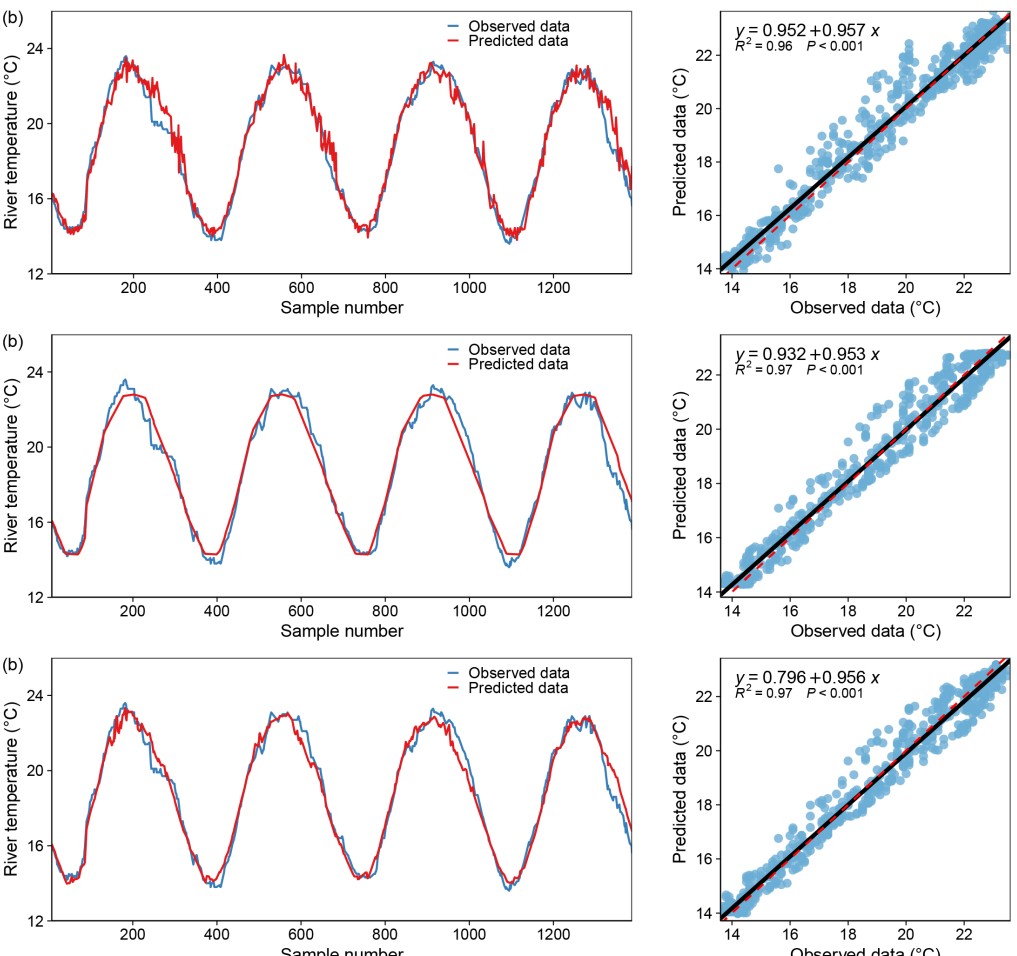

**Figure 12.** Model fitting results—Multilayer Perceptron Neural Network, blue dot: X coordinate (observed data), Y coordinate (predicted data) ; black line: y = x ; red dotted line: the regression curve of the blue dots. (**a**) only one input variable (DOY), (**b**) with two input variables (DOY and Flow), (**c**) with all variables.

For the importance ranking of the six selected machine learning models (DT, RF, GB, AB, SVR, MLPNN), Figures 4–6 showed an interesting result that DOY (day of the year) is the most influential input variable. Furthermore, DOY had a much greater influence on predictions than the second and third-ranked variables, i.e., discharge and AT (air temperature). The influence of DOY on the water temperature can be explained in two aspects. First, DOY allowed the ML models to reproduce the annual variability of the water temperature [47]. DOY is not only a time series but also a proxy representing the current as well as the historically cumulative impacts of the upstream reservoir operation. As it can be seen from Figure 2, the variation of the water temperature in the cascaded reservoirs showed a clear annual cycle pattern, i.e., the inter-annual variation of the water temperature does not vary drastically. For natural rivers, AT is typically the critical factor

influencing water temperature [48]. However, it can be seen from Figure 2 that there is a significant lag in water temperature on the time scale relative to the variation in AT. This water temperature delay phenomenon can be attributed to the operation of the upstream reservoirs, which also explains why air temperature is less important in influencing water temperature predictions.

Second, less obviously, the DOY allows the model to include the seasonal variability of stratification, which is an important feature of reservoirs with deep water [49]. Specifically, the water temperature in the surface layer of the reservoir (epilimnion) is warmer; it varies greatly and is strongly influenced by the meteorological conditions. In the middle transient layer (metalimnion), the reservoir water temperature shows a drastic gradient change in the vertical direction. While the water temperature at the bottom layer of the reservoir (hypolimnion) is colder and varies uniformly [50]. The release gate of the reservoir is usually built at the bottom of the reservoir, which means that there is a difference between the water temperature released from the reservoir and the water temperature in the downstream rivers. Generally, the water temperature released from the reservoir is lower than that of the downstream in summer and vice versa in winter [51–53]. Water released from the reservoir, therefore, leads to a more stable intra-annual variation of river water temperature. It also explains that the discharge shown is a critical factor in predicting water temperature.

More dams will be built and planned in the upper reaches of the Yangtze River, which will further influence water temperature, affecting fish spawning. Therefore, the accurate prediction of water temperature can better guide reservoir operation in order to reduce the ecological impact of reservoir construction on the Yangtze River. Our results provide evidence that it may be feasible to use adaptive management strategies aimed at restoring the environmental flow of the river that allows to adequately mimic natural river water temperature [54,55]. To this end, it is necessary to continuously monitor river flows and air temperatures downstream of dams and to facilitate the modeling of future scenarios based on modified reservoir outflows.

## 5. Conclusions

Water temperature is a critical indicator of the river's overall health. Accurate temperature prediction is a critical aspect of river management. We evaluated six models and found that all of them perform well in terms of prediction accuracy. In contrast to the majority of earlier machine learning applications, which simply used air temperature as a predictor, we utilized a variety of inputs. The results indicated that GB and RF models outperformed all other models with lower root mean square error and higher $R^2$ and NSE values. The DOY is the most significant factor in all forecast models, while air temperature, flow and dew temperature are secondary aspects connected to water temperature variations. This is possibly due to the cascaded reservoir operating on a year-round basis that affects the annual cycle of downstream river water temperature. As a result, the DOY may be more precise in its prediction of river water temperature downstream of the reservoir; the modeling performance indicates that the machine learning model developed in this study is capable of accurately predicting river water temperature under the influence of a cascaded reservoir.

**Author Contributions:** Conceptualization, D.J. and Y.X.; methodology, Y.L.; software, K.W.; validation, K.W.; formal analysis, J.G.; data curation, J.G.; writing—original draft preparation, K.W.; writing—review and editing, Y.X. and D.J.; supervision, Y.X. and Y.L. All authors have read and agreed to the published version of the manuscript.

**Funding:** This research was funded by the Research funding of China Three Gorges Corporation (Grant No. 202003251), the National Key Research and Development Program of China (Grant No. 2019YFE0109900) and the National Natural Science Foundation of China (Grant No. U2040205, 52109013).

**Institutional Review Board Statement:** Not applicable.

**Informed Consent Statement:** Not applicable.

**Data Availability Statement:** The data used in this study are available on request from the corresponding author.

**Acknowledgments:** The constructive comments and suggestions of the anonymous reviewers and the editors are gratefully acknowledged.

**Conflicts of Interest:** The authors declare no conflict of interest.

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
