# Peer review of "Forecasting Water Temperature in Cascade Reservoir Operation-Influenced River with Machine Learning Models"

_water, doi:10.3390/w14142146_

Round 1
Reviewer 1 Report
The authors have done a good job of writing a concise and generally easy to follow manuscript, which facilitates more helpful reviews. I have two major concerns that will need to be addressed regarding the novelty and application of the study, and the statistical methods. These intersect and I have to ask what the relevance of using these models to predict water temperature really is, when simpler methods seem to suffice. The authors need to convince me there is a benefit to using these more complicated methods over just using air temperature. That is, I want to explicitly see the improvements gained, and I want to know if they are even meaningful – given the paper is supposed to be about predicting temperature of fish spawning sites, are the differences in results with simpler methods meaningful.
I have two “medium” comments, the title invokes the importance to fish spawning sites, yet this is never discussed after the results. This connection needs to be clearer in the introduction and discussed later or removed from the title. Second, there seemed to be a significant amount of methods that appeared in the results; the study is not replicable from the methods alone.
I also noticed some minor errors that I have pointed out below, in addition to being more specific with my major and medium concerns.
First, I want the authors to understand my train of thought, which hopefully allows them to understand why I’m asking for a change and they can either make the change or clarify what they did if I am mistaken.
I am not familiar with most of the statistical methods used, but I immediately wondered what happened when you used any of these with highly correlated variables. In linear regression, multicollinearity can cause significant interpretation problems, with parameter estimates switching signs depending on what else is in the model; there are numerous ways of dealing with this including using just one variable if another is highly correlated. I started with the random forest, and found this:
http://blog.datadive.net/selecting-good-features-part-iii-random-forests/
My take from this is that if day of year and air temperature are highly correlated, then the random forest model is always going to suggest the second one is far less important. Reading through the methods, there is no mention of what the model used actually was (or the software). One has to skip to the results to figure out what the variables in the model were. So my question is, how much of an improvement are any of these models over just using air temperature and/or regression models? And, how does having multiple highly correlated variables affect model performance or interpretations?
Second, I wasn’t sure what a “cascade reservoir” might be as it’s not defined, so I googled it and found this: https://www.mdpi.com/2073-4441/11/5/1008/htm
Which predicted water temperature from air temperature, in the same study area, and published in the same journal. Yet, not cited in this manuscript. So while I know what the term “cascade reservoir” is, I am now wondering why this relevant study was not mentioned. I would still suggest defining cascade reservoir though.
Line by line comments:
Take fish spawning sites out of the title, or else improve the discussion and make it relevant/worthy of including in the title.
21 – What is “river health” and why are you citing a methods paper to support this statement? Cite a paper that presents data on temperature and “river health.” This is lazy citation – go to Qui et al., read the references there, cite those references that are applicable. Do not give credit to Qui et al. for work they didn’t do, but only cited themselves. Caissie (2006) is a review paper and a much better starting point for the paragraph.
30-35: this is a two sentence paragraph, a practice that is generally best avoided in scientific writing. Combine with preceding or following paragraphs.
51: physics or physical-based?
62: Check language/grammar in this sentence, “machine learning models differs”
67: Italicize Latin names
82: in this case, you need to define “accurately” How much better does 3 factors have to be over 2 factors to be “accurate?” This whole approach seems like it needs to follow an AIC type framework
86-87: this would be a good place to expand on how it relates to fish spawning sites, if you want to keep it in the title
Figure 1 – I have no idea where in the world this is, please include a global locator map. Also, you should probably include a scale bar. Captions need to make the figure stand alone.
108: Another 2-sentence paragraph, which uses abbreviations that have not yet been introduced. Remove or expand this.
Somewhere in the methods you need to describe the models for each method and the software used.
174: SVM not defined
199-202: One sentence paragraph, it’s not necessary to tell us what you are about to do and then do it. The sub-headings are enough.
Table 1: This table is nowhere near stand-alone. Nothing in the caption is defined.
208: “As shown in Table 1” is poor writing. Write a sentence on the results you want to show, and reference the table to draw attention to the data supporting your statement, e.g., start this sentence with “All six models…” this also applies elsewhere in the manuscript (figure 6 call out in the text, for example).
209: Give a citation for the Nash efficiency coefficient
216-226: These are methods, not results
Incidentally, for RF as an example, I want to see how well DOY compares to just air temperature, not nested sets of models. That’s a more fair comparison.
Figure 3: make this caption stand-alone (here and elsewhere)
Most of the results are quite tedious to read. I am not sure how to help, other than to tell the authors you are probably losing a lot of readers here because the reading is dense. Consider making it shorter and really focus on the important parts.
389: So…. What’s your take-home message? Use whatever model you want? Also, hard to make any real conclusions if none of your models were just air temperature. How much of an improvement did you see over just air temperature? If you did this, then I’m just not seeing it in the results.
Author Response
Thanks to the editor and reviewers, all comments have been accepted and revised. Please see the attachment.

Reviewer 2 Report
Comment 1: The subject addressed is within the scope of the journal.
Comment 2: The major defect of this study is the debate or Argument is not clear stated in the introduction session. Hence, the contribution is weak in this manuscript. I would suggest the author to enhance your theoretical discussion and arrives your debate or argument.
Comment 3: Especially, the introduction section needs to re-organize. The major debate or Argument is not clear stated in the introduction session. Hence, the contribution debates are weak in this manuscript. I would suggest the author to enhance your literature discussion and arrives your debate or argument.
Comment 4: I would like to request the author to emphasis on the contributions on practically and academically in implication session.
Comment 5: Methods section determines the results. Kindly focus on three basic elements of the methods section.
a. How the study was designed?
b. How the study was carried out?
c. How the data were analyzed?
Comment 6: Please explain your results into steps and links to your proposed method.
Comment 7: It is suggested to add articles entitled “Ekwueme & Agunwamba. Trend Analysis and Variability of Air Temperature and Rainfall in Regional River Basins”, “Kumar & Singh. A Comparison between MLR, MARS, SVR and RF Techniques: Hydrological Time-series Modeling” and “S. S. Ojha et al. Comparison of Meteorological Drought using SPI and SPEI” to the literature review.
Comment 8: Please make sure your conclusions' section underscore the scientific value added of your paper, and/or the applicability of your findings/results, as indicated previously. Please revise your conclusion part into more details. Basically, you should enhance your contributions, limitations, underscore the scientific value added of your paper, and/or the applicability of your findings/results and future study in this session.
Comment 9: The discussion section needs to be described scientifically. Kindly frame it along the following lines:
i. Main findings of the present study
ii. Comparison with other studies
iii. Implication and explanation of findings
iv. Strengths and limitations
v. Conclusion, recommendation, and future direction.
Author Response

(The authors gave the same response as above.)

Reviewer 3 Report
Thank you for submitting your manuscript to the Water journal. Generally, thetopic fits into the scope of the journal, and the structure respects Scientific
Best Practice. However, the content requires revision. In the literature review, it is important that the scientific novelty of the work is established through a critical analysis of related literature. In the current version, the literature review needs improvement. A substantial international review of the relevant peer reviewed literature
must be performed and must be the base of that section. With this, followng
questions must be clarified: How does the present work contribute towards the
gaps identified? How does it improve upon previous work? It is recommended
that a short discussion of the novel contribution of each reference cited
shall be provided to give readers a better understanding of their relevance. Thus, the main questions of the reviewer are: What is the scientific motivation
for the study? Which scientific question shall be answered with this? What is
your scientific hypothesis that you wish to answer with the inbestigation?
Is there a scientific need for the comparison of machine learning methods.
Putting the scientific motivation will also help you to identify the novelties
that characterises a scientific publication. The methodology shuld be improved. I strongly recommend to include a flow chart illustrating the steps of the methodology in the beginning of the methodology section. Moreover, there is a need to include an overview
on the comparison criteria for the models, and why those criteria have been
selected. Also sections 2.2.1 to 2.2.6 should be supported with peer review
scientific references. Section 4 should be entiteled Results, and should contain only results. A discussion section must be added and filled with reasonable content. In the conclusions, in addition to summarising the actions taken and results, please strengthen the explanation of their significance. It is recommended to use quantitative reasoning comparing with appropriate benchmarks, especially those stemming from previous work.
Author Response

(The authors gave the same response as above.)

Round 2
Reviewer 1 Report
I believe the authors have satisfactorily addressed my concerns. However, because of the way the changes are presented in the text, it is very difficult to read (not the fault of the authors).
Author Response
Thank you
Reviewer 2 Report
All comments have been addressed properly, so the article suggested to publish with its present form.
Author Response
Thank you for your review.
Reviewer 3 Report
My comments have been considered
Author Response
Thank you for your review